# Different cell imaging methods did not significantly improve immune cell image classification performance

**Taisaku Ogawa[1], Koji Ochiai[2], Tomoharu Iwata[3], Tomokatsu Ikawa[4], Taku Tsuzuki[5,6], Katsuyuki Shiroguchi[1]\*, Koichi Takahashi[2]\***

**1** Laboratory for Prediction of Cell Systems Dynamics, RIKEN Center for Biosystems Dynamics Research (BDR), Suita, Osaka, Japan, **2** Laboratory for Biologically Inspired Computing, RIKEN Center for Biosystems Dynamics Research (BDR), Suita, Osaka, Japan, **3** Ueda Research Laboratory, NTT Communication Science Laboratories, Kyoto, Japan, **4** Division of Immunology and Allergy, Research Institute for Biomedical Sciences, Tokyo University of Science, Noda, Chiba, Japan, **5** Laboratory for Computational Molecular Design, RIKEN Center for Biosystems Dynamics Research (BDR), Suita, Osaka, Japan, **6** Epistra Inc., Tokyo, Japan

\* katsuyuki.shiroguchi@riken.jp (KS); ktakahashi@riken.jp (KT)

**Data Availability Statement:** A minimal data set is available on figshare at https://doi.org/10.6084/m9.figshare.14789811.v1 (https://figshare.com/articles/dataset/Minimum_dataset_for_Different_cell_imaging_methods_did_not_significantly_

## Abstract

Developments in high-throughput microscopy have made it possible to collect huge amounts of cell image data that are difficult to analyse manually. Machine learning (e.g., deep learning) is often employed to automate the extraction of information from these data, such as cell counting, cell type classification and image segmentation. However, the effects of different imaging methods on the accuracy of image processing have not been examined systematically. We studied the effects of different imaging methods on the performance of machine learning-based cell type classifiers. We observed lymphoid-primed multipotential progenitor (LMPP) and pro-B cells using three imaging methods: differential interference contrast (DIC), phase contrast (Ph) and bright-field (BF). We examined the classification performance of convolutional neural networks (CNNs) with each of them and their combinations. CNNs achieved an area under the receiver operating characteristic (ROC) curve (AUC) of ~0.9, which was significantly better than when the classifier used only cell size or cell contour shape as input. However, no significant differences were found between imaging methods and focal positions.

## Introduction

Recent advances in automated microscopy have made it possible to collect large numbers of cell images [1]. However, it is becoming increasingly difficult to analyse these large amounts of data manually. Therefore, the automation of judgements by machine learning is expected to improve the speed and processing of large amounts of data and ensure consistency in the results of judgements [2, 3]. The application of machine learning to biological image processing is expanding with the development of deep learning [4–6]. An interesting use of deep learning in cell biological research is to infer the differentiation of living cells. For example,

improve_immune_cell_image_classification_
performance_/14789811). The entire training data
set is over 500GB and cannot be made public, but
it may be available by contacting the authors.

**Funding:** This work was supported by a project
subsidised by the RIKEN Junior Research
Associate program (to K. O.), the Special
Postdoctoral Researchers Program of RIKEN (to T.
O.) and MEXT KAKENHI Grant Number 18H05411
(K.S.). Epistra, Inc. provided support in the form of
salaries for authors Taku Tsuzuki and Koichi
Takahashi, but did not have any additional role in
the study design, data collection and analysis,
decision to publish, or preparation of the
manuscript. The specific roles of these authors are
articulated in the 'author contributions' section.

**Competing interests:** Epistra, Inc. provided
support in the form of salaries for authors Taku
Tsuzuki and Koichi Takahashi, but did not have any
additional role in the study design, data collection
and analysis, decision to publish, or preparation of
the manuscript. This commercial affiliation does
not alter our adherence to PLOS ONE policies on
sharing data and materials.

Buggenthin et al. predicted lineage choice in differentiating primary hematopoietic progenitors from image patches of brightfield microscopy and cellular movement [7]. However, the impact of cell observation methods on cell type identification on machine learning-based cell information extraction has not been systematically investigated.

In this study, we examined the classification performance of convolutional neural networks (CNNs), a representative network architecture, in cell image classification of the lymphoid-primed multipotential progenitor (LMPP) and pro-B cells. LMPP cells differentiate into pro-B cells in the process of differentiation of hematopoietic cells to B cells. We obtained cell images using three different types of imaging methods and their combinations: differential interference contrast (DIC), phase contrast (Ph), and bright-field (BF). We then evaluated the dependence of classification performance on the imaging methods using the area under the curve (AUC) of the receiver operating characteristic (ROC) as a performance measure. AUCs were examined for different focal positions and different numbers of cell images given to the CNN for training. Our results indicated that the effects of the imaging methods on the AUC were not significant, and the AUC was ~0.9, for most cases.

This paper is organised as follows. First, we explain how cell images are acquired, processed, learned and evaluated. Next, we discuss how the method of photography and the number of samples affect the classification performance.

## Materials and methods

The workflow of the experiment consisted of six steps, as shown in Fig 1.

### Sample preparation

Induced leukocyte stem cells (iLS cells) were used to prepare two different immune states of CD19$^-$ LMPP and CD19$^+$ pro-B cells [8]. The LMPP state of the iLS cells was maintained on Tst-4 stromal cells in the presence of 10 ng/mL IL-7 (R&D Systems, USA), stem cell factor (R&D), Flt-3L (R&D) and 40 nM 4-OHT (Sigma-Aldrich, USA). The pro-B state was induced by culturing the cells on Tst-4 cells with 5 ng/mL IL-7 in the absence of 4-OHT for 7 days.

### Observation dishes

Both LMPP and pro-B cells were stained with 5 μg/mL Alexa Fluor 594 (Alx594)-conjugated anti-CD19 antibody fluorescence (115552; BioLegend) and sorted using a fluorescence-activated cell sorter (FACS Aria III; BD) using CD19$^-$ and CD19$^+$ gates. After sorting, the same numbers of LMPP and pro-B cells were mixed, diluted with Iscove's Modified Dulbecco's Medium (IMDM) (Thermo Fisher Scientific, USA) at a cell density of $4.7 \times 10^5$ cells/mL and seeded on three glass-bottom dishes (Eppendorf, Germany) for Experiment 1 and 2, respectively. We note that both LMPP and pro-B cells sorted here contained multiple phases of the cell cycle.

### Microscopy

All cell images were acquired using a Nikon Eclipse Ti2 inverted microscope with a 40 × water immersion objective lens (CFI Apo Lambda S 40XC WI; Nikon, Japane) and captured with a scientific CMOS camera (Zyla 5.5; Andor, UK). BF and DIC images were observed with Nikon's standard units, and Ph images were observed with an external phase contrast unit (Ti2-T-BP-E; Nikon). Alx594-conjugated anti-CD19 antibody fluorescence images were observed with standard epifluorescence optics consisting of an LED (pE-300 white; CoolLED,

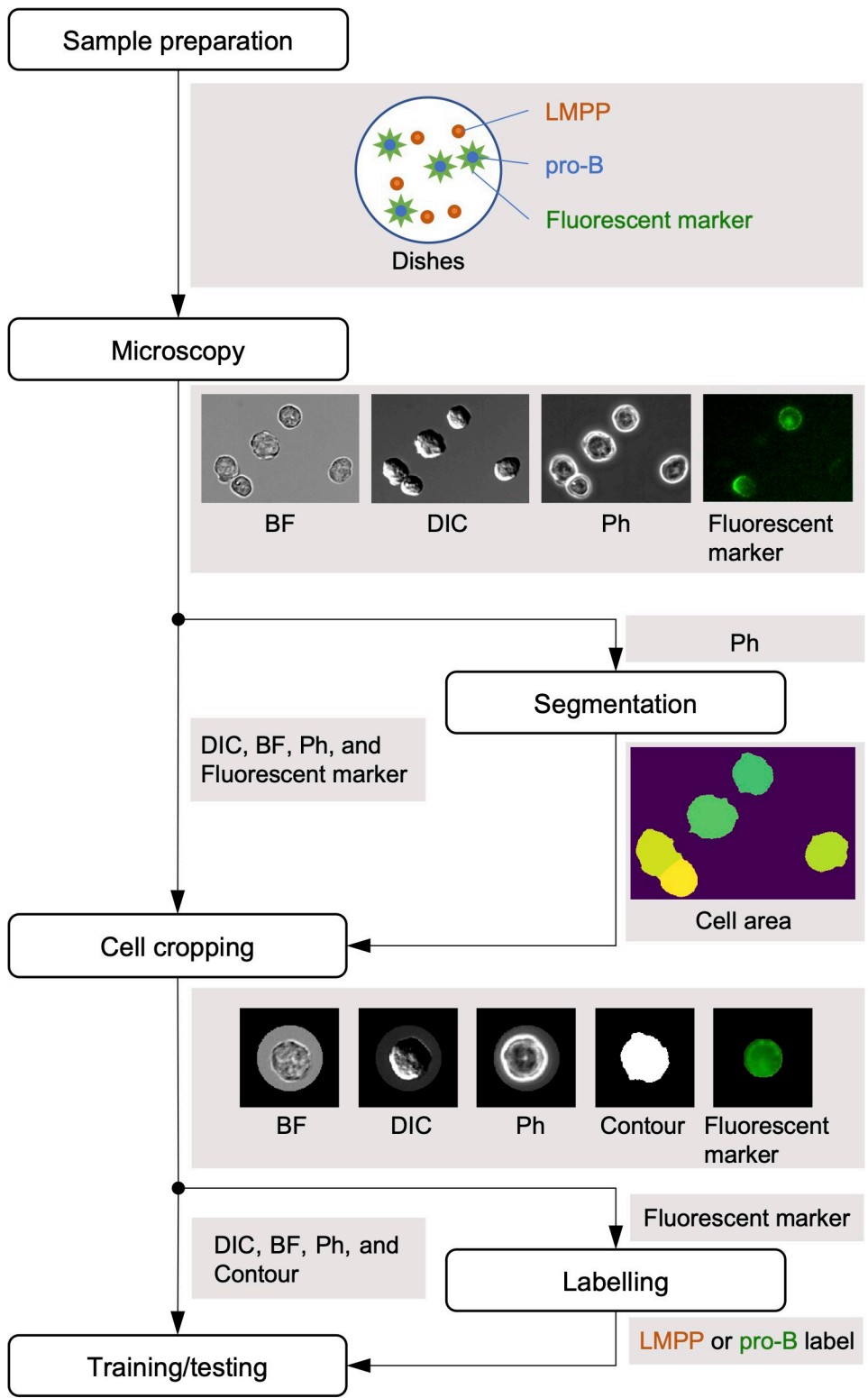

**Fig 1. Workflow of the experiment.** The experiment consisted of six processes: sample preparation, microscopy, segmentation, cell cropping, labelling and training/testing. To standardise imaging conditions, we mixed LMPP and pro-B at sample preparation. On microscopy, images were obtained in four channels: DIC, BF, Ph and Alexa Fluor 594 (Alx594)-conjugated anti-CD19 antibody fluorescence. We cut out individual cells based on Ph images. Next, cells were labelled based on the intensity of the Alx594-conjugated anti-CD19 antibody fluorescence. Finally, training and testing were applied to cell images to evaluate classification performance.

UK) and a mirror unit (mCherry HQ; Nikon). Exposure times were 10 ms for BF, DIC and Ph, and 100 ms for Alx594.

All image acquisition procedures were controlled by Nikon software (NIS-element). To examine which is the best focal position for the performance of the CNN, Z-stacks (from -3.6 μm to +2.7 μm, 0.3-μm interval) of Alx594, BF, DIC and Ph were sequentially recorded and repeated for 144 fields of view (410 μm × 346 μm/field) (Experiment 1). The home focal position (0 μm in Z-stacks) was determined by observation of 56 cells in the first field of view of the first dish; a focal position where the area of most cells in Alx594-conjugated anti-CD19 antibody fluorescence images were apparently (by eyes) the largest was defined as the home focal position. This home focal position was recorded using Nikon Perfect Focus System (PFS) (TI2-N-ND-P), and used for all three dishes in Experiment 1. Focal position was initially moved to the home focal position before taking Z-stacks of each channel using PFS, and subsequently moved from—3.6 μm to + 2.7 μm with 0.3 μm interval by the motorized focusing unit of Ti2-E microscope. Furthermore, to investigate whether continuous shots of the same image channel and focal position improved the performance, one shot of Alx594 and thirty continuous shots (3 s) of BF, DIC and Ph were recorded sequentially and repeated for 204 fields of view (Experiment 2). The focal position was determined and recorded as Experiment 1. The focal position was moved to the home focal position before taking each shot of each channel using PFS. Induction of pro-B state and preparation of the observation dishes were performed differently in experiments 1 and 2.

## Segmentation

Cell regions were extracted from Ph images. Details regarding the extraction procedures are presented in Fig 2. First, the Otsu method [9] of automatic threshold determination was applied to an original phase-contrast microscopy image (Fig 2A) to determine the intensity threshold. Next, the image was binarised using the threshold and morphological closing was applied to the binarised image [10, 11]. The results are shown in Fig 2B. The regions touching the border were then removed [12] (Fig 2C) and holes were filled [13] (Fig 2D).

As shown in Fig 2D, regions contained multiple cells when cells were touching. To separate the cells, the watershed method was applied [14]. The watershed marker was determined by the following procedure. First, Euclidean distance transform [15] and Gaussian filter [16] were applied, as shown in Fig 2E. Next, the pixels of local peaks were used as markers [17].

The boundaries of regions after applying watershed are shown in Fig 2F. This result still included large regions containing multiple cells or small regions containing no cells. These regions were removed from the train/test data by removing regions $\leq$ 2000 pixels and $\geq$ 8000 pixels. For all microscopic images in this study, pixel resolution was 162.5 nm/pixel was used, so 2000 pixels and 8000 pixels correspond to 52.81 μm$^2$ and 211.2 μm$^2$, respectively.

## Cropping

Images of 150 × 150 pixels were cropped from BF, DIC, Ph and Alx594-conjugated anti-CD19 antibody fluorescence images for each cell in the image. We also cropped cell regions themselves as binarised images. In addition, we calculated cell sizes as the total number of pixels in the cell region. To reduce the effects of background intensity, the Alx594-conjugated anti-CD19 antibody fluorescence images were divided by the intensity of the blurred Alx594-conjugated anti-CD19 antibody fluorescence images before cropping. The blurred image was generated by applying a gaussian filter with sigma = 200[pixel].

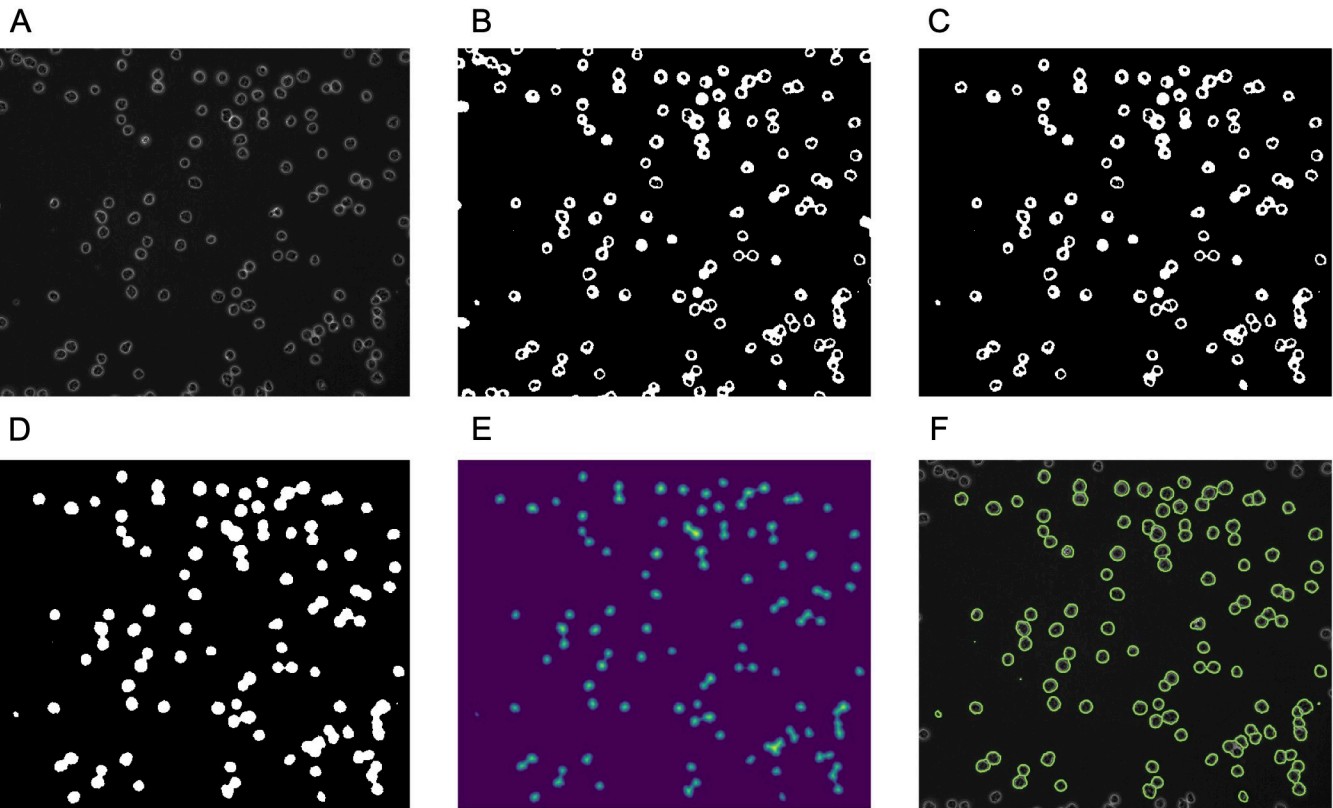

**Fig 2. Workflow of cell segmentation.** (A) Original phase-contrast microscopy image. (B) Binary image. (C) Binary image after removing the regions touching the border. (D) Hole filling result. (E) Euclidean distance transform result. (F) Boundaries of cell regions superimposed on the original image.

## Labelling

To identify cell types, the Gaussian mixture method was applied to the intensity histograms of Alx594-conjugated anti-CD19 antibody fluorescence. The intensity of the $i$th cell image in an experiment was calculated by:

$$F_i = \log_e\left(\frac{l_i}{\max(\mathbf{l})}\right) \tag{1}$$

where $l_i$ is an element of $\mathbf{l}$ and represents the total intensity of pixels in the region < 30 pixels from the centre of the cell image.

Cells were labelled as LMPP, pro-B or unused by applying a two-component Gaussian mixture model to the histogram. Cells with a probability < 80% in both categories were labelled as unused and were not used for CNN training and testing (Fig 3A). The range of unused was $(-2.652 < F_i < -2.376)$ in Experiment 1. In the rest of the cells, the high- and low-intensity groups were labelled as pro-B and LMPP, respectively.

## Training/testing

In the last step, training and testing were conducted and the AUC was calculated. Training and testing were performed in a cross-validation manner. Cross-validation is a method to evaluate a model, which separates data into training data and test data. A model is fitted to the training data and evaluated by using the test data. These training and evaluation procedures

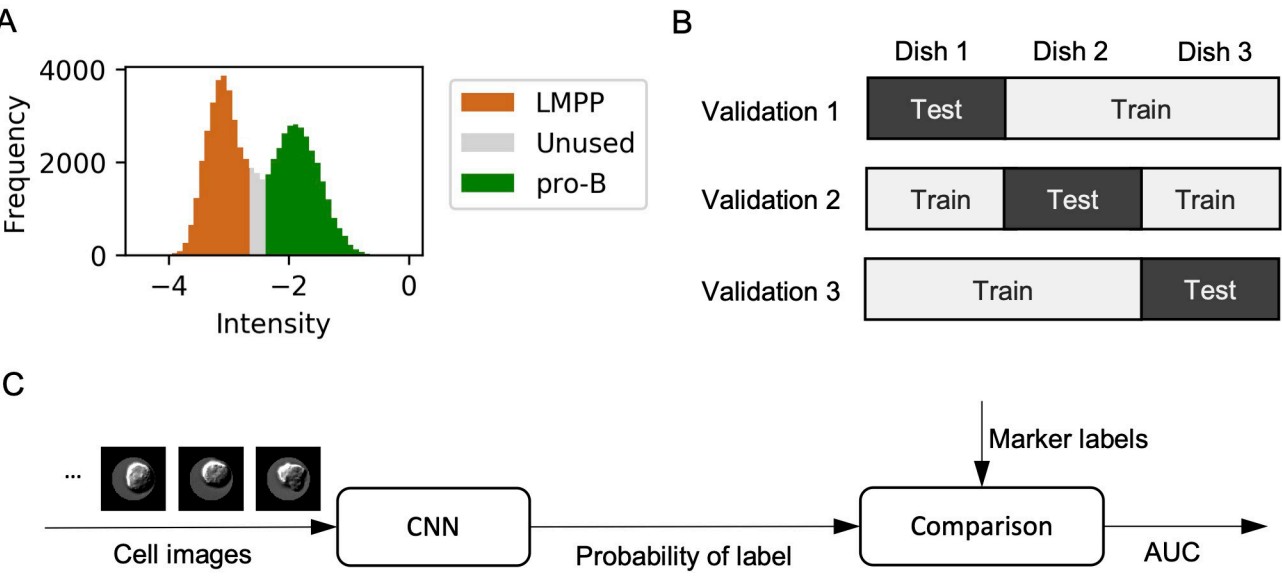

**Fig 3. Image data preparation and cross-validation scheme.** (A) Labelling: natural logarithm of intensity histogram. The intensity was calculated by Eq (1). LMPP, pro-B and unused are colour-coded as brown, green and grey, respectively. (B) Training/testing: cross-validation with the dish as the unit. Images from the $n$th dish were used for testing in the $n$th validation. (C) Schematic of the test process in training and testing. Cell images were input into the CNN to obtain the label probability. The AUC was calculated from these label probabilities and corresponding marker labels.

were repeated while changing the test data. Cell images from dishes 1, 2 and 3 were used as test data in validations 1, 2 and 3, respectively. Fig 3B shows the cross-validation matrix of dish columns and validation rows.

The training/testing processes consisted of three steps: pre-processing, training of the deep neural network and evaluation. These steps are described in detail in the following subsections.

**Pre-processing.** Three image processing methods were applied to cell images before feeding them to the deep neural network. 1) Cell images were masked with a circle of radius 50 pixels from the centre and the pixels outside the mask were set to 0. The reason why we did not use the mask created from the phase contrast is to avoid the influence of the phase contrast information on other channels through the mask shape. 2) For normalisation, the average intensity of pixels inside the mask was adjusted to 1. 3) In addition, the cell number for labels in each dish was trimmed by a random choice to Eq (2) where $n$ is dish number, $t$ is cell type (LMPP or pro-B) and $C_{n,t}$ is the number of cells of type $t$ in dish $n$. The numbers of cells in Experiment 1 are shown in Table 1. The numbers of cells were reduced twice, i.e., in labelling and pre-processing procedures. The numbers of cells in Experiment 2 are

**Table 1. Number of cells in Experiment 1.**

|  | Experiment 1 |
|---|---|
| **Number of cells cropped** | 62,891 |
| **Number of cells labelled** | 57,589 |
| **Number of cells used for training** | 57,252 |

The number of cells in each image processing step in Experiment 1; the table in Experiment 2 is shown in supporting information.

shown in S1 Table in S1 File.

$$\min_{n,t}(C_{n,t}) \tag{2}$$

**Training and testing of CNN and SVM.**   A CNN or support vector machine (SVM) was used to identify cell types, i.e., LMPP and pro-B. The CNN had four convolution layers (16 channel kernel size 5, 16 channel kernel size 5, 32 channel kernel size 5, 32 channel kernel size 5, but when using images with $\geq 2$ channels, the numbers of channels were all doubled) and two fully connected layers (2,000 nodes, 2 nodes). ReLU (see Eq (3)) was used as the non-linear activation function in all hidden layers. The CNN architecture is identical to Xu's study [6], except that we changed parameters regarding the resolution and number of channels of the images to fit the dataset we used. In Eq (3), $x$ is the output of the prior layer. The softmax function (see Eq (4)) was applied at the output of the neural network, and here $x_i$ is the $i$th node of output layer $x$. A max-pooling layer with kernel size 3 and slide 2 was placed behind each convolution layer. A drop rate of 0.1 dropout layer was added in all gaps in the layers.

$$ReLU(x) = \max(x,\ 0) \tag{3}$$

$$softmax(\boldsymbol{x}) = \frac{e^{x_i}}{\sum_{j=1}^{N} e^{x_j}} \tag{4}$$

The following training methods were applied to this CNN. To prevent over-fitting, an early stopping method was used during training. In this study, 10% of training data was used for validation. The CNN was fitted to the remaining 90% of the data, and the loss for validation data was calculated at the end of each epoch. If the loss of the validation data did not decrease for five consecutive epochs, the learning was stopped and the model with the smallest loss for validation was used for evaluation.

Binary cross-entropy was used for the loss function, as defined by Eq (5), where $y$ is the label and $\hat{y}$ is the categorical probability CNN output. It becomes minimum when $y = \hat{y}$. Adam, a method for learning rate optimisation [18], was used for learning with parameters $\alpha = 0.0001$, $\beta1 = 0.9$, $\beta2 = 0.999$, $\varepsilon = 10^{-8}$ and $\eta = 1.0$. The batch size was 32. Learning was stopped if the epoch achieved 30 regardless of loss change.

$$H(y,\ \hat{y}) = -\mathrm{y}\log(\hat{y}) - (1-y)\log(1-\hat{y}) \tag{5}$$

Fig 3C shows the workflow of the classification performance test. When using SVM, cell size was used as the input and the value of the decision function was used to calculate the AUC. Kernel of SVM is radial basis function and kernel coefficient parameter is 1.

## Results

This section examines cell identification classification performances with different imaging and classification methods.

We compared classification by cell contour using CNN and classification by cell size using SVM. The AUC was higher for classification by cell contour than by cell size. Classification performance of the CNN with single-channel images (DIC, BF or Ph) was evaluated. All of BF, Ph and DIC showed better performances than both SVM with cell size and CNN with only cell image contours (Fig 4A).

We then used stacked multiple images (DIC + BF, DIC + Ph, BF + Ph, DIC + BF + Ph) as input for the CNN. In all combinations tested, we found no significant differences in AUC

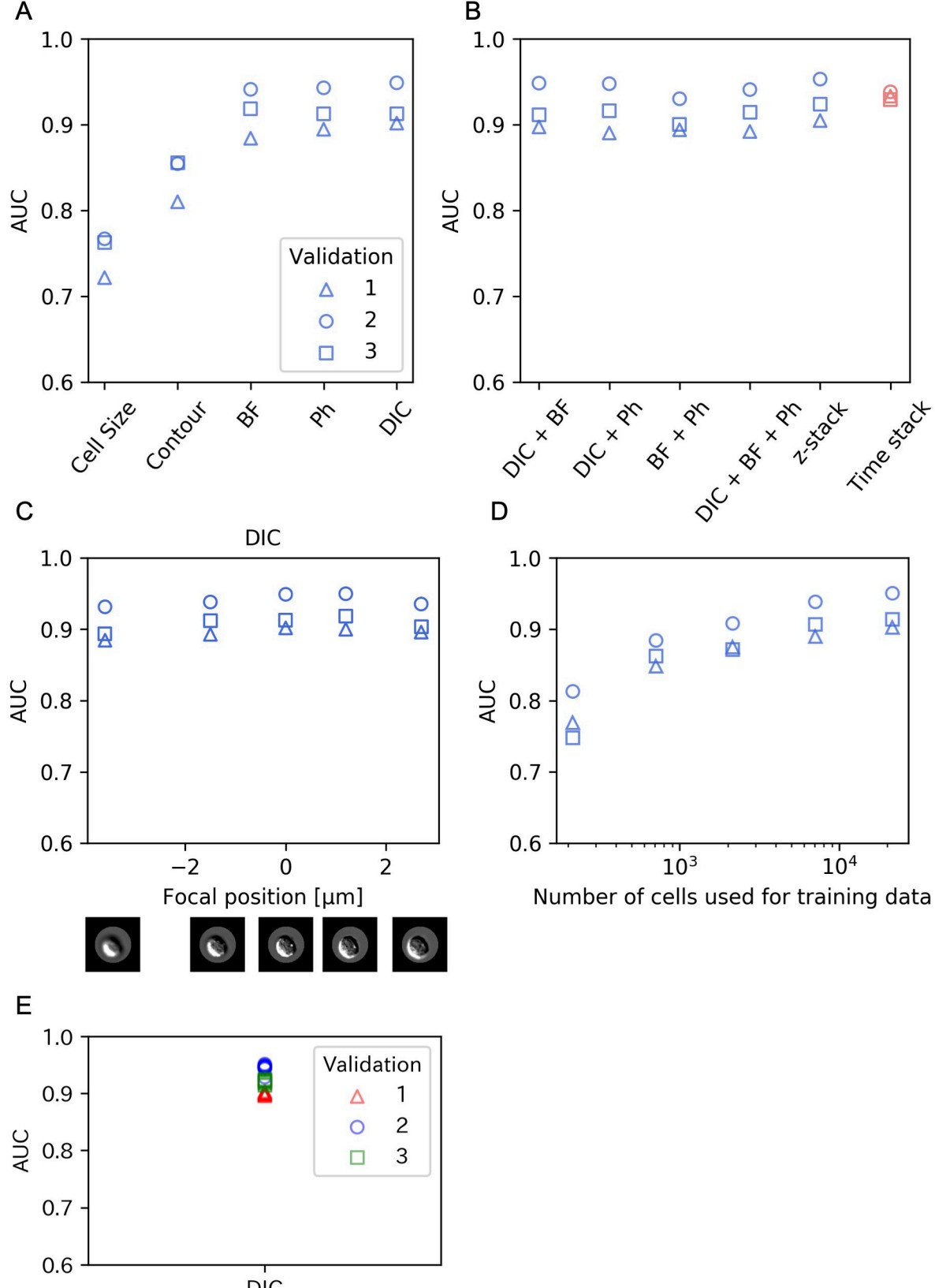

**Fig 4. Experimental results from cross-validation.** In all figures, triangles, circles and squares represent validation 1, 2 and 3, respectively. In addition, blue and red marks represent data from experiments 1 and 2, respectively. (A) Cross-validation results for SVM input: cell size and CNN inputs: contour and single-channel (BF, Ph, DIC). (B) Cross-validation results for CNN inputs: combination of multiple channels, z- and time stacks of DIC. (C) AUCs are shown as a function of focal positions. The shot focus was −3.6, −1.5, 0.0, 1.2 and 2.7 μm. Zero is the reference position calibrated with the Alx594-conjugated anti-CD19 antibody fluorescence image. (D) The AUCs in DIC images are shown as a function of the cell sample numbers in CNN training while the size of the test was fixed at 10,722 cell images. (E) Variation of DIC depending on the initial random number of CNN learning.

from DIC single-channel results (Fig 4B). We also tested z-stacking and time stacking, and neither yielded obvious improvement of performance. This trend was maintained in Experiment 2 (S1 Fig in S1 File). Cross-validation while shifting the focus on DIC images resulted in no significant degradation of the AUC in the range examined (from—3.6 μm to + 2.7 μm) (Fig 4C). The same results were found for BF and Ph (S2 Fig in S1 File).

To examine how the classification performance changes with the size of training data, we created data sets with different numbers of cell images (214; 708; 2,144; 7,077 and 21,444 cells) by random sampling. The AUC increased with increasing number of cells contained in the training data, but the performance gain by adding more samples tended to diminish as the data size became larger (Fig 4D).

Sometimes, CNN performance can be affected by the initial values of the parameters. To confirm that our results were unaffected by this effect, we trained the network with different initial random numbers. Variations originating from different initial values were smaller than variations in classification performance between dishes (Fig 4E) in DIC images.

## Discussion

In this study, we checked the classification performance of CNN by the following five investigations: 1) comparison of the AUC between the cases using only cell size or cell shape and where cell images were given as input; 2) the AUC when multi-channel inputs with imaging methods were given; 3) variation of the AUC when shifting the focus; 4) variation of the AUC when changing the amount of training data; and 5) variation of the AUC depending on the initial random number.

The CNN showed better classification performance when BF, Ph or DIC images were given than when using only cell size or image contours. This result suggests that CNN uses information derived from the internal structure of the cells reflected in the texture of cell images, and not only the shapes or sizes of the cells.

No significant degradation in AUC was observed over the focal range examined.

With regard to the dataset size, an AUC of ~0.8 was maintained even at 708 cells. This suggests that CNN can be used even when a relatively small dataset is available depending on the purpose. However, testing of the network would be a challenge in such cases (in this study, we used a fixed testing dataset size of 10,722 cells).

When stacked multiple images (DIC + BF, DIC + Ph, BF + Ph, DIC + BF + Ph) were used as input for the CNN, no significant improvement of performance over single-channel results was found.

The performance of the CNN seemed to be less affected by the optical microscopy technique and focus than the variance between dishes. Further investigations regarding the generality of this observation are necessary. For example, closer analyses on whether this property is maintained for other cells, whether it is maintained even when the number of classes is increased to > 2, and whether it is maintained even when other architectures (such as ResNet) are applied, would convey useful insights. In this study, however, we showed that if a suitable network architecture is chosen and imaging experiments were properly designed and

conducted (e.g the number of cells, image preprocessing etc.), the effects of some important factors including imaging methods and focal position on classification accuracy can be limited or negligible, at least in the classification task between LMPP and pro-B cells.

## Supporting information

**S1 File.**
(DOCX)

## Acknowledgments

We thank M. Watabe for English correction and K. Nishida and K. Fukuhara for technical support.

## Author Contributions

**Conceptualization:** Katsuyuki Shiroguchi, Koichi Takahashi.

**Data curation:** Taisaku Ogawa.

**Funding acquisition:** Katsuyuki Shiroguchi, Koichi Takahashi.

**Investigation:** Taisaku Ogawa.

**Methodology:** Tomoharu Iwata.

**Project administration:** Katsuyuki Shiroguchi, Koichi Takahashi.

**Resources:** Tomokatsu Ikawa.

**Software:** Koji Ochiai, Taku Tsuzuki.

**Supervision:** Katsuyuki Shiroguchi, Koichi Takahashi.

**Validation:** Koji Ochiai.

**Writing – original draft:** Taisaku Ogawa, Koji Ochiai.

**Writing – review & editing:** Katsuyuki Shiroguchi, Koichi Takahashi.

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
