## [Decision Letter · Decision Letter 0]

12 Jan 2021

PONE-D-20-33334

Different cell imaging methods did not significantly improve immune cell image classification performance

PLOS ONE

Dear Dr. Takahashi,

Thank you for submitting your manuscript to PLOS ONE. After careful consideration, we feel that it has merit but does not fully meet PLOS ONE’s publication criteria as it currently stands. Therefore, we invite you to submit a revised version of the manuscript that addresses the points raised during the review process.

We look forward to receiving your revised manuscript.

Kind regards,

Chi-Hua Chen, Ph.D.

Academic Editor

PLOS ONE

Journal Requirements:

3.

Thank you for stating the following in the Competing Interests: "NO authors have competing interests"

We note that one or more of the authors are employed by a commercial company:  Epistra Inc, Japan.

(2) Please also provide an updated Competing Interests Statement declaring this commercial affiliation along with any other relevant declarations relating to employment, consultancy, patents, products in development, or marketed products, etc.  

Reviewers' comments:

Reviewer's Responses to Questions

**Comments to the Author**

1. Is the manuscript technically sound, and do the data support the conclusions?

Reviewer #1: Partly

Reviewer #2: Yes

Reviewer #3: Partly

2. Has the statistical analysis been performed appropriately and rigorously? 

Reviewer #1: Yes

Reviewer #2: No

Reviewer #3: Yes

3. Have the authors made all data underlying the findings in their manuscript fully available?

Reviewer #1: Yes

Reviewer #2: Yes

Reviewer #3: Yes

4. Is the manuscript presented in an intelligible fashion and written in standard English?

Reviewer #1: Yes

Reviewer #2: Yes

Reviewer #3: Yes

5. Review Comments to the Author

Reviewer #1: I do not find enough merit to support the publication of the paper, as the contribution is not significant enough.

The authors do not compare the proposed approach with actual state of art. Several methods are proposed in the literature.

It is not clear the novelty of the method.

Are not clear the pros and cons of the proposed approach respect the actual state of the art.

Overall impression of study is just below average; the study could be useful and deals with significant problem, but the style way in which paper is written does not match the journal standards.

Reviewer #2: Different cell imaging methods did not significantly improve immune cell image

classification performance

Summary:

This paper studies the use of three microscopy imaging modalities on accuracy of image-based cell classification. The three imaging modalities are: differential interference contrast (DIC), phase contrast (Ph) and bright-field (BF). The cell types are lymphoid-primed multipotential progenitor (LMPP) and pro-B cells. The cell classification methods are custom-designed convolutional neural networks (CNNs) and support vector machine (SVM) using cell size and cell contour shape as inputs to SVM. The ground truth cell classification is established using Alexa Fluor 594 stain, fluorescent imaging, and simple image analysis.

The conclusions of this study are: (1) the accuracy of image-based cell classification is invariant to the three microscopy imaging modalities and their combinations, (2) the CNN classifiers outperform the SVM classifiers, and (3) the accuracy of image-based cell classification is invariant to a chosen focal position. The conclusions are supported by three replicates (three dishes) and multiple spatial fields of views (FOVs) and multiple time points (time lapse FOVs) in two experiments.

Major comments:

• The authors should include a clear description of experimental and observational factors and their levels in this experimental design study. Right now, the reader cannot easily derive such critical information to this study.

o For example, it is not clear whether the cells are going through mitosis during time lapse imaging and therefore the cell state is ignored as a factor.

o The focal position is listed as a factor. The levels of this factor are captured as z-stacks (± 3.0 �m range, 0.3-�m interval). First, the authors should explain how they samples the focal positions (e.g., frequently occurring deviation from a perfect focus, visually indistinguishable image appearance). Second, the claim about classification accuracy being invariant to focal position should be limited to the chosen range (otherwise it might not be true).

• The authors should explain why they used a custom designed CNN architecture as opposed to using a widely used U-Net CNN architecture.

o The authors could include a picture of the custom-designed CNN architecture.

o If the authors are not using the U-Net, then they could compare the CNN architectural designs (custom vs U-Net)

• The authors should include confidence intervals in their reported results.

o For example, “CNN showed the best classification performance with DIC images (AUC ~0.9).” on page 15. How do we know that the classification results obtained using DIC images are statistically any better than the results obtained by the other imaging modalities?

Minor comments:

• Page 8: “These regions were removed from the train/test data by removing regions ≤ 2000 pixels and ≥ 8000 pixels”. Can you provide physical dimensions accompanied by some apriori knowledge?

• Page 9: “…fluorescence images were divided by the intensity of the blurred Alx594-conjugated ..” Who defines which frame in a z-stack is blurred?

• Page 11: “Cell images were masked with a circle of radius 50 pixels …” Why are the cell images masked with a circle if you have a segmentation mask? I would also recommend clarify the terminology for raw cell images, mask images from segmentation (ground truth), and mask images created by inserted circles into background color.

• Page 16, line 258: “different numbers of cell images (214, 708, 2,144, 7,077 and 21,444 cells)” Please, use semicolons to separate the numbers if you are using commas to separate thousands. Include the numbers of cell images for CNN training into a list of factors explored.

• The authors claim that the data are available, but there is no URL pointing to the data.

Reviewer #3: In this work, the authors studied the effects of deep learning-based cell classification using images from several common bright field imaging methods. These imaging methods include differential interference contrast (DIC), phase contrast, and regular bright field. They used lymphoid-primed multipotential progenitor (LMPP) and pro-B cells as their model system. Their results showed that the performance of Deep learning classifier from different imaging methods performed similarly. Overall, this is an interesting study, and the results suggested that regular bright-field images without contrast optics (i.e., phase, DIC) can provide sufficient information for Deep learning. However, the authors only demonstrate this result in one biological model system (LMPP vs. proB) with one simple deep learning architecture. Hence, it is not clear if the reported findings represent a general theme or restricted to this very system that the authors demonstrated in the study. As it is well known that the performance of deep learning is highly associated with network architectures. Thus, it is not clear if or how different network architecture (such as ResNet, google net, mobile net …etc.) of deep learning can affect the classification performance in recognizing different types of biological specimens/cell samples under different imaging modalities. Overall, more thorough studies should be conducted to better support their claims.

6. PLOS authors have the option to publish the peer review history of their article (what does this mean?). If published, this will include your full peer review and any attached files.

Reviewer #1: No

Reviewer #2: **Yes: **Peter Bajcsy

Reviewer #3: No

---

## [Author Response · Author response to Decision Letter 0]

25 Jun 2021

Please see 'Responses to Reviewers-rev.docx', which includes some additional data and figures.

---

## [Decision Letter · Decision Letter 1]

22 Jul 2021

PONE-D-20-33334R1

Different cell imaging methods did not significantly improve immune cell image classification performance

PLOS ONE

Dear Dr. Takahashi,

Thank you for submitting your manuscript to PLOS ONE. After careful consideration, we feel that it has merit but does not fully meet PLOS ONE’s publication criteria as it currently stands. Therefore, we invite you to submit a revised version of the manuscript that addresses the points raised during the review process.

We look forward to receiving your revised manuscript.

Kind regards,

Chi-Hua Chen, Ph.D.

Academic Editor

PLOS ONE

Reviewers' comments:

Reviewer's Responses to Questions

**Comments to the Author**

1. If the authors have adequately addressed your comments raised in a previous round of review and you feel that this manuscript is now acceptable for publication, you may indicate that here to bypass the “Comments to the Author” section, enter your conflict of interest statement in the “Confidential to Editor” section, and submit your "Accept" recommendation.

Reviewer #1: (No Response)

Reviewer #2: All comments have been addressed

Reviewer #3: (No Response)

2. Is the manuscript technically sound, and do the data support the conclusions?

Reviewer #1: No

Reviewer #2: Yes

Reviewer #3: No

3. Has the statistical analysis been performed appropriately and rigorously? 

Reviewer #1: No

Reviewer #2: Yes

Reviewer #3: N/A

4. Have the authors made all data underlying the findings in their manuscript fully available?

Reviewer #1: Yes

Reviewer #2: No

Reviewer #3: Yes

5. Is the manuscript presented in an intelligible fashion and written in standard English?

Reviewer #1: Yes

Reviewer #2: Yes

Reviewer #3: Yes

6. Review Comments to the Author

Reviewer #1: I do not find enough merit to support the publication of the paper, as the contribution is not

significant enough.

your tests are not sufficient to validate your conclusions. It is not clear the novelty of the method.

Are not clear the pros and cons of the proposed approach respect the actual state of the art.

Overall impression of study is just below average; the study could be useful and deals with

significant problem, but the style way in which paper is written does not match the journal

standards.

Reviewer #2: Different cell imaging methods did not significantly improve immune cell image

classification performance

Minor comments:

• Line 94: “three glass-bottom dishes (Eppendorf, Germany). For” – remove the period

• Line 95: “Experiment 1 and 2, respectively.” – Table 1 includes description of Experiment 1. I cannot find anywhere information about Experiment 2. Can you clarify?

• Previous comment: “The authors claim that the data are available, but there is no URL pointing to the data.”

o Response: “The data will be made available from the corresponding author upon reasonable request. Since the size of the datasets is larger than 1 TB, it is difficult to put it on a public place with a URL.”

o Recommendation: add a sentence that summarizes the data size reaching 1TB as a multiplication of image pixel size x number of fields of view taken per dish x number of dishes x etc.

• Previous comment: “The authors should explain why they used a custom designed CNN architecture as opposed to using a widely used U-Net CNN architecture.

The authors could include a picture of the custom-designed CNN architecture.

If the authors are not using the U-Net, then they could compare the CNN architectural designs (custom vs U-Net)”

o Response:” Our CNN network is a plain CNN (no special connections such as skip connections and attention are added), and only changes made were on network parameters such as kernel sizes. Parameters used are given in the section ‘Training and testing of CNN and SVM’

(Main text lines 210-213).

As far as we understand, U-Net is a network for image transformation and is not often used for identification. Although the main focus of this study is to investigate the effect of different observation methods on the accuracy of identification, it is an interesting question whether our findings hold for other architectures (such as ResNet). We have added this point to Discussion (Main text lines 296-302).”

o Recommendation: add a sentence that relates “your definition of a plain CNN” (i.e., there is no published definition of a plain CNN) to a well-known published CNN architecture. For example, you should state how your CNN architecture is different from AlexNet or LeNet and why you made those changes.

Reviewer #3: The reviewer's comments were not directly addressed in the revised version of the manuscript. The main claim authors attempting to establish is that regular bright-field images without contrast optics (i.e., phase, DIC) can provide sufficient information for Deep learning. Yet, this was only demonstrated in one biological model system (LMPP vs. proB) with one simple deep learning architecture. Thus, I still don’t think the present data provide sufficient evidence to support the claim.

7. PLOS authors have the option to publish the peer review history of their article (what does this mean?). If published, this will include your full peer review and any attached files.

Reviewer #1: No

Reviewer #2: **Yes: **Peter Bajcsy

Reviewer #3: No

---

## [Author Response · Author response to Decision Letter 1]

3 Sep 2021

Responses to Reviewers’ Concerns for Taisaku Ogawa et al., “Different cell imaging methods did not significantly improve immune cell image classification performance”

We would like to thank all the reviewers again for reading through the manuscript and giving insightful and constructive comments.

We have further refined our manuscript. All changes we made are highlighted in yellow in the revised manuscript. 

Reviewer #1

Concern #1

I do not find enough merit to support the publication of the paper, as the contribution is not

significant enough. your tests are not sufficient to validate your conclusions. It is not clear the novelty of the method. Are not clear the pros and cons of the proposed approach respect the actual state of the art. Overall impression of study is just below average; the study could be useful and deals with significant problem, but the style way in which paper is written does not match the journal standards.

Thank you for carefully considering the revised version of our manuscript. Our tests were designed according to the purpose of this study, that is to investigate the impact of using different observation methods and settings on cell classification accuracy, and not to exhibit the performance of a novel image analysis method. We agree that this study deals with a significant problem, and believe that many experimentalists and data scientists working in the area of cell imaging would find our results useful in designing and deciding their experiments and data analysis pipelines. 

We are thankful to Reviewer #1’s critical readings and comments, which we found very useful. However, some of the comments were a bit abstract and ambiguous, therefore, what the reviewer expects us to improve the quality of the manuscript is not very clear -- (e.g. ‘Overall impression of study is just below average’, ‘the style way in which paper is written does not match the journal standards’). We would very much appreciate it if some more elaboration on the reviewer’s intent and more concrete comments on his/her concerns.

Reviewer #2

Concern #1

• Line 94: “three glass-bottom dishes (Eppendorf, Germany). For” – remove the period

• Line 95: “Experiment 1 and 2, respectively.” – Table 1 includes description of Experiment 1. I cannot find anywhere information about Experiment 2. Can you clarify?

Deleted the period in line 94. 

The number of cells in experiment 2 is listed in the supplement information. We added that to the caption of table 1.

Concern #2

Previous comment: “The authors claim that the data are available, but there is no URL pointing to the data.”

o Response: “The data will be made available from the corresponding author upon reasonable request. Since the size of the datasets is larger than 1 TB, it is difficult to put it on a public place with a URL.”

o Recommendation: add a sentence that summarizes the data size reaching 1TB as a multiplication of image pixel size x number of fields of view taken per dish x number of dishes x etc.

A part of the dataset is now available at the following URL.

https://figshare.com/articles/dataset/Minimum_dataset_for_Different_cell_imaging_methods_did_not_significantly_improve_immune_cell_image_classification_performance_/14789811

1TB is the total size of the original images. The cropped images were used for training. 0.6% of the original images are now available at the URL above. The cropped images were excluded from the public data because they can be generated from the original images using the method described in the paper.

Concern #3

Previous comment: “The authors should explain why they used a custom designed CNN architecture as opposed to using a widely used U-Net CNN architecture.

The authors could include a picture of the custom-designed CNN architecture.

If the authors are not using the U-Net, then they could compare the CNN architectural designs (custom vs U-Net)”

o Response:” Our CNN network is a plain CNN (no special connections such as skip connections and attention are added), and only changes made were on network parameters such as kernel sizes. Parameters used are given in the section ‘Training and testing of CNN and SVM’

(Main text lines 210-213).

As far as we understand, U-Net is a network for image transformation and is not often used for identification. Although the main focus of this study is to investigate the effect of different observation methods on the accuracy of identification, it is an interesting question whether our findings hold for other architectures (such as ResNet). We have added this point to Discussion (Main text lines 296-302).”

o Recommendation: add a sentence that relates “your definition of a plain CNN” (i.e., there is no published definition of a plain CNN) to a well-known published CNN architecture. For example, you should state how your CNN architecture is different from AlexNet or LeNet and why you made those changes.

We provided the information on the CNN architecture in Main text (lines 212-). As the purpose of our work is not to propose a novel network architecture, we think that our description contains enough information to reproduce our work. However, we added a citation to the paper by Xu et al (2017), in which readers can find the exact architecture ours is based on.

Reviewer #3

Concern #1

The reviewer's comments were not directly addressed in the revised version of the manuscript. The main claim authors attempting to establish is that regular bright-field images without contrast optics (i.e., phase, DIC) can provide sufficient information for Deep learning. Yet, this was only demonstrated in one biological model system (LMPP vs. proB) with one simple deep learning architecture. Thus, I still don’t think the present data provide sufficient evidence to support the claim.

Thank you for carefully considering the revised version of our manuscript. 

‘The main claim authors attempting to establish is that regular bright-field images without contrast optics (i.e., phase, DIC) can provide sufficient information for Deep learning.’

 -> Yes, it constitutes one of our main claims.

‘Yet, this was only demonstrated in one biological model system (LMPP vs. proB) with one simple deep learning architecture.’

 -> Yes, it was our experimental design. We found that at least the relatively simple deep learning architecture we used attains acceptable classification accuracy (i.e. AUC >~ 0.9), regardless of some important factors including imaging modalities and focal position,.

‘Thus, I still don’t think the present data provide sufficient evidence to support the claim.’

 -> We disagree. Our tests were designed according to the purpose of this study, that is to investigate the impact of using different observation methods and settings on cell classification accuracy (and, not to exhibit the performance of a novel image analysis method). We investigated whether it is *possible* to design a simple experimental setup and data analysis pipeline without expensive optics and complicated DL architecture for cell image classification, if such optical setups can provide sufficient information. Logically, this is a yes/no question. Therefore, just one set of clear and statistically sound positive results should convey useful information. 

 With that said, we very much agree with the reviewer that more thorough studies should be conducted to further investigate the dependence of the classification performance on different cell types, optics and network architectures. It will be an important follow-up to the current study. In the previous revision, we made this point clear in main text lines 296-303.

---

## [Decision Letter · Decision Letter 2]

8 Nov 2021

PONE-D-20-33334R2Different cell imaging methods did not significantly improve immune cell image classification performancePLOS ONE

Dear Dr. Takahashi,

Thank you for submitting your manuscript to PLOS ONE. After careful consideration, we feel that it has merit but does not fully meet PLOS ONE’s publication criteria as it currently stands. Therefore, we invite you to submit a revised version of the manuscript that addresses the points raised during the review process.

We look forward to receiving your revised manuscript.

Kind regards,

Chi-Hua Chen, Ph.D.

Academic Editor

PLOS ONE

Journal Requirements:

Reviewers' comments:

Reviewer's Responses to Questions

**Comments to the Author**

1. If the authors have adequately addressed your comments raised in a previous round of review and you feel that this manuscript is now acceptable for publication, you may indicate that here to bypass the “Comments to the Author” section, enter your conflict of interest statement in the “Confidential to Editor” section, and submit your "Accept" recommendation.

Reviewer #1: All comments have been addressed

Reviewer #2: (No Response)

2. Is the manuscript technically sound, and do the data support the conclusions?

Reviewer #1: Yes

Reviewer #2: Yes

3. Has the statistical analysis been performed appropriately and rigorously? 

Reviewer #1: Yes

Reviewer #2: Yes

4. Have the authors made all data underlying the findings in their manuscript fully available?

Reviewer #1: Yes

Reviewer #2: No

5. Is the manuscript presented in an intelligible fashion and written in standard English?

Reviewer #1: Yes

Reviewer #2: Yes

6. Review Comments to the Author

Reviewer #1: Revision well done, the authors have addressed concerns of all the reviewers improving the submitted paper.

Reviewer #2: Minor comments:

From the rev 1 and rev 2:

+++++++++++++

Concern #2

Previous comment: “The authors claim that the data are available, but there is no URL

pointing to the data.”

o Response: “The data will be made available from the corresponding author upon

reasonable request. Since the size of the datasets is larger than 1 TB, it is difficult to

put it on a public place with a URL.”

o Recommendation: add a sentence that summarizes the data size reaching 1TB as a

multiplication of image pixel size x number of fields of view taken per dish x number of

dishes x etc.

A part of the dataset is now available at the following URL.

https://figshare.com/articles/dataset/Minimum_dataset_for_Different_cell_imaging_meth

ods_did_not_significantly_improve_immune_cell_image_classification_performance_/14

789811

1TB is the total size of the original images. The cropped images were used for

training. 0.6% of the original images are now available at the URL above. The cropped

images were excluded from the public data because they can be generated from the

original images using the method described in the paper.

+++++++++++++++++

The authors claimed that “A part of the dataset is now available at ..”. However, The URL just shows a list of file folders (dish1, dish3, dish2) and a list of file names. There is no way to download or view the images.

From rev1 and rev2:

+++++++++++++++++++

Concern #3

Previous comment: “The authors should explain why they used a custom designed

CNN architecture as opposed to using a widely used U-Net CNN architecture.

The authors could include a picture of the custom-designed CNN architecture.

If the authors are not using the U-Net, then they could compare the CNN architectural

designs (custom vs U-Net)”

o Response:” Our CNN network is a plain CNN (no special connections such as skip

connections and attention are added), and only changes made were on network

parameters such as kernel sizes. Parameters used are given in the section ‘Training

and testing of CNN and SVM’

(Main text lines 210-213).

As far as we understand, U-Net is a network for image transformation and is not often

used for identification. Although the main focus of this study is to investigate the effect

of different observation methods on the accuracy of identification, it is an interesting

question whether our findings hold for other architectures (such as ResNet). We have

added this point to Discussion (Main text lines 296-302).”

o Recommendation: add a sentence that relates “your definition of a plain CNN” (i.e.,

there is no published definition of a plain CNN) to a well-known published CNN

architecture. For example, you should state how your CNN architecture is different

from AlexNet or LeNet and why you made those changes.

We provided the information on the CNN architecture in Main text (lines 212-). As the

purpose of our work is not to propose a novel network architecture, we think that our

description contains enough information to reproduce our work. However, we added a

citation to the paper by Xu et al (2017), in which readers can find the exact architecture

ours is based on.

+++++++++++++++

I agree that your description contains enough information about the changes of the UNet architecture. My comment was about explaining why you had to make those changes instead of using the published UNet as is. This comment takes into account your description presented at the lines 211-219:

+++++++++++++++++++++++++++

211 The CNN architecture is based on Xu’s study [6]. The CNN had four convolution

212 layers (16 channel kernel size 5, 16 channel kernel size 5, 32 channel kernel size 5, 32 channel

213 kernel size 5, but when using images with ≥ 2 channels, the numbers of channels were all

214 doubled) and two fully connected layers (2,000 nodes, 2 nodes). ReLU (see Eq (3)) was used

215 as the non-linear activation function in all hidden layers. In Eq (3), x is the output of the prior

216 layer. The softmax function (see Eq (4)) was applied at the output of the neural network, and

217 here xi is the ith node of output layer x. A max-pooling layer with kernel size 3 and slide 2

218 was placed behind each convolution layer. A drop rate of 0.1 dropout layer was added in all

219 gaps in the layers.

+++++++++++++++++++++++

7. PLOS authors have the option to publish the peer review history of their article (what does this mean?). If published, this will include your full peer review and any attached files.

Reviewer #1: No

Reviewer #2: No

---

## [Author Response · Author response to Decision Letter 2]

14 Dec 2021

Reviewer #2

Concern #1: No further comments raised.

Concern #2

"The authors claimed that “A part of the dataset is now available at ..”. However, The URL just shows a list of file folders (dish1, dish3, dish2) and a list of file names. There is no way to download or view the images."

Response: When you open the figshare.com URL below, you see the list of folders and the list of files, but these files are not clickable (weird, but it is how figshare.com founders designed it – probably because they store the data in .zip format only.). Instead, please scroll down and find the red ‘Download’ button next to the ‘Cite’ button, which allows you to download the dataset (3.08GB).

https://figshare.com/articles/dataset/Minimum_dataset_for_Different_cell_imaging_methods_did_not_significantly_improve_immune_cell_image_classification_performance_/14789811

Concern #3: 

"I agree that your description contains enough information about the changes of the UNet architecture. My comment was about explaining why you had to make those changes instead of using the published UNet as is. This comment takes into account your description presented at the lines 211-219:."

Response: Our architecture is identical to Xu’s Deep CNN (we do not use UNet), except that we modified a few parameters regarding the resolution and the number of channels.

We added a sentence to the manuscript to make this point clearer.

---

## [Editor Report · Decision Letter 3]

23 Dec 2021

Different cell imaging methods did not significantly improve immune cell image classification performance

PONE-D-20-33334R3

Dear Dr. Takahashi,

We’re pleased to inform you that your manuscript has been judged scientifically suitable for publication and will be formally accepted for publication once it meets all outstanding technical requirements.

Kind regards,

Chi-Hua Chen, Ph.D.

Academic Editor

PLOS ONE
---

## [Editor Report · Acceptance letter]

30 Dec 2021

PONE-D-20-33334R3 

Different cell imaging methods did not significantly improve immune cell image classification performance 

Dear Dr. Takahashi:

I'm pleased to inform you that your manuscript has been deemed suitable for publication in PLOS ONE. Congratulations! Your manuscript is now with our production department. 

Kind regards, 

on behalf of

Professor Chi-Hua Chen 

Academic Editor

PLOS ONE